# Headache in Children: Selected Factors of Vascular Changes Involved in Underlying Processes of Idiopathic Headaches

**DOI:** 10.3390/children7100167

**Published:** 2020-10-04

**Authors:** Joanna Sordyl, Ewa Małecka-Tendera, Beata Sarecka-Hujar, Ilona Kopyta

**Affiliations:** 1Department of Pediatrics and Pediatric Endocrinology, Faculty of Medical Sciences in Katowice, Medical University of Silesia, 40-752 Katowice, Poland; etendera@sum.edu.pl; 2Department of Basic Biomedical Sciences, Faculty of Pharmaceutical Sciences in Sosnowiec, Medical University of Silesia in Katowice, 41-200 Sosnowiec, Poland; beatasarecka@poczta.onet.pl; 3Department of Pediatric Neurology, Faculty of Medical Sciences in Katowice, Medical University of Silesia, 40-752 Katowice, Poland; ilonakopyta@autograf.pl

**Keywords:** atherosclerosis, headache, idiopathic headache, inflammation, child, dyslipidemia, pediatric obesity

## Abstract

Headaches are common complaints in children. The International Classification of Headache Disorders, 3^rd^ edition (beta version), defines more than 280 types of headaches. Primary headaches refer to independent conditions that cause pain and include migraine, tension-type headaches (TTH), and trigeminal autonomic cephalalgias (TACs). Several agents are involved in the pathogenesis of headaches. The factors associated with predisposition to atherosclerosis seem to be particularly important from the clinical point of view. The influence of obesity on the incidence of headaches has been well established. Moreover, idiopathic headaches, especially migraine, are thought to be one of the first signs of disorders in lipid metabolism and atherosclerosis. The risk of migraine increases with increasing obesity in children. Another factor that seems to be involved in both obesity and headaches is the adiponectin level. Recent data also suggest new potential risk factors for atherosclerosis and platelet aggregation such as brain-derived neurotrophic factor (BDNF), sCD40L (soluble CD40 ligand), serpin E1/PAI I (endothelial plasminogen activator inhibitor), and vascular endothelial growth factor (VEGF). However, their role is controversial because the results of clinical studies are often inconsistent. This review presents the current knowledge on the potential markers of atherosclerosis and platelet aggregation, which may be associated with primary headaches.

## 1. Introduction

Headache is one of the most frequent complaints in primary care practices and a very common condition reported by children, adolescents, and young adults [1,2,3,4]. However, headache phenotypes differ depending on the age group. In the pediatric population, the incidence of headaches increases with age. Conicella et al. observed headaches in 66% of school-age children [5]. The authors demonstrated that 93% of the analyzed children presented a recent onset of headache, whereas medium- and late-onset headaches were reported by 3% and 4% of all patients, respectively [5]. The overall headache prevalence has been reported to be as high as 56% in children under 10 years of age and 91% in early adulthood [3,6]. Additionally, a relationship between gender and the frequency of headaches was observed in pediatric patients. Abu-Arafeh et al. described odds ratio (OR) equal to 1.53 and 1.67 for the prevalence of headache and migraine in female and male patients, respectively [3,6]. During puberty, headaches are more common in girls, whereas in prepuberty, the frequency of headaches is similar in males and females [6,7]. On the other hand, in a large group of adult patients from the USA, migraine was more common among females, but in the case of other severe headaches, no sex difference was observed [2,8]. In some patients, headaches may become a long-lasting health problem as they tend to persist into adulthood. Most patients from Nova Scotia, Canada, having pediatric headaches suffered from headaches in adulthood (i.e., twenty years after diagnosis), whereas 27% were headache-free [9].

Headache classification is a complex issue due to the multi-faceted pathophysiology of this disorder. Possible etiological factors include, inter alia, neuronal, vascular, immunological, and psychological agents [10]. Nowadays, more than 280 headache types and subtypes are defined and categorized in the International Classification of Headache Disorders, 3^rd^ edition, created by the International Headache Society (ICHD-3). The classification consists of four sections: primary headaches, secondary headaches, neuropathies, and appendix [10].

The group of primary headaches includes migraine, tension-type headaches (TTH), trigeminal autonomic cephalalgias (TACs), and other primary headache disorders [10].

The diagnosis of primary headache is strongly related to the patient‘s medical history and typical characteristics of headaches described in ICHD-3. A thorough medical history and physical examination are crucial for identifying the etiology [3,10,11].

The treatment of headaches in children and adolescents requires a balanced approach to personalize therapy depending on the kind, frequency, and severity of symptoms as well as the limitations in daily life activities due to pain. The main goal of headache therapy in children is quick resolution with minimal side effects. The multi-faceted effort includes three general stages: lifestyle modification, psychotherapy, and pharmacotherapy [12,13].

A number of risk factors for headache have been demonstrated. Predisposition to atherosclerosis in idiopathic headaches seems to be particularly important from the clinical point of view. Migraine was suggested to be one of the first signs of disorders in lipid metabolism [14]. Elevated lipid concentrations were widely observed in adult migraineurs as well as in a few clinical studies conducted in pediatric patients [14,15,16]. The risk of migraine was also suggested to increase with increasing obesity in children [17,18,19,20].

Recent data suggest that some risk factors for atherosclerosis and platelet aggregation (i.e., brain-derived neurotrophic factor (BDNF), sCD40L (soluble CD40 ligand), serpin E1/PAI I (endothelial plasminogen activator inhibitor), and vascular endothelial growth factor (VEGFO)) may be involved in the underlying processes of idiopathic headaches [21,22,23,24,25,26,27,28,29,30].

The knowledge on the correlation between headaches and factors, which may increase the risk of atherosclerosis, and consequently, premature coronary artery disease, may be of particular importance, especially in the pediatric population.

The aim of the present literature review was to present current knowledge on the role of potential markers of atherosclerosis and platelet aggregation that may be associated with primary headaches. Our aim was to focus on the pediatric population. However, due to the small amount of data on this topic, we also discussed the results of studies conducted in groups of adults.

## 2. Methodology

A literature search was conducted to identify reports on the relationship between BMI and dyslipoproteinemia with primary headaches as well as on the new potential biomarkers of vascular changes and platelet aggregation that may be associated with primary headache pathophysiology. The material was collected systematically over several months (the last update: 19 September 2020). The search was conducted using PubMed, Embase, Google Scholar, Scopus, and SciELO databases. The following keywords and MeSH terms were used in different combinations: “primary headaches”, or “idiopathic headache”, or “headache disorders”, or “tension headache”, or “migraine disorders”, and “child”, or “pediatrics”, or “children”, and “obesity”, or “body mass index”, or “BMI”, or “dyslipoproteinemia”, or “atherosclerosis”, or “vascular changes”, or “platelet aggregation”, or “brain-derived neurotrophic factor”, or “BDNF”, or “soluble CD40 ligand”, or “sCD40L”, or “endothelial plasminogen activator inhibitor”, or “serpin E1/PAI-1”, or “vascular endothelial growth factor”, or “VEGF”, or “adiponectin”.

The inclusion criteria for the paper were as follows: (a) abstracts and titles of publications relevant to the topic; (b) peer-reviewed publications in English; and c) research involving human subjects. Studies were excluded if they focused on secondary headaches, drug intervention, and when they were conducted in patients with accompanying diseases that may have potentially affected the selected biomarkers and contributed to the symptoms described by patients (e.g., autoimmune disease, epilepsy, mental disorders). The authors decided to focus particularly on original studies performed in children. The data obtained from adults as well as reviews were used only in the case of insufficient scientific pediatric reports. Particular attention was given to the articles published within the last five years. Earlier data were only used for their substantive value.

The example search process in Embase using all the keywords allowed us to identify a total of 179,464 articles. After the initial screening, 77,077 papers other than full-length articles were removed, leaving 102,387 articles. After another search, papers written only in English were included with a total number of 95,102 articles. Clinical studies in humans concerned 30,918 articles, of which 3342 were performed in children (from newborns to adolescents). Then, the search was performed separately for each of the factors that we aimed to describe in the present study (using combinations of the selected keywords). The same search process was used for the other databases. A total of 1242 articles were included in further analysis. The titles and abstracts of these papers were assessed by two independent authors, and those irrelevant to the topic or duplicates were removed. In addition, the reference lists of the reviewed publications were also searched. For all remaining papers, the full text of the papers was read to determine whether relevant information was included. Eventually, the search yielded 118 publications that were included in the present review.

## 3. Results

### 3.1. Body Mass Index (BMI)

The available data support the relationship between obesity and headaches in children and adolescents [17,18,19,20,31,32,33,34,35,36,37,38,39,40,41,42]. Previous data demonstrated that obesity is associated with more frequent headaches in children and adolescents [17,18,19,20,38,39,40,41]. According to Pinhas-Hamiel et al., obese girls had a 4-fold increase in headaches compared to lean controls [17]. In the adult population, Peterlin et al. found that obese individuals had an 81% greater risk of episodic migraine than subjects with normal weight. The risk was particularly high in younger adults (below 50 years of age) and in females [38].

BMI percentile correlated not only with the frequency of headaches, but also with headache-related disability. However, no correlation between obesity and headache severity and duration was found [18,20]. The body mass index of children with headaches correlates not only with the incidence of this ailment, but also with the tendency to relapse and everyday functioning disruptions [18]. Verotti et al. reported that a change of mean BMI from 32.9 kg/m^2^ at baseline to 29.9 kg/m^2^ after 12 months of an intervention program caused a significant reduction in the headache frequency in obese Italian children and adolescents [43]. Similar data were obtained in the study evaluating the obesity–headache relationship in adolescents (13–18 years). Obese adolescent girls and boys were more likely to have recurrent headaches [43]. In contrast, some authors have reported no correlation between migraine and increased BMI in children [40].

Data on the relationship between obesity and TTH in children are insufficient and controversial, and the obtained results are often contradictory [19,39,40,41]. Pinhas-Hamiel et al. suggested that slightly overweight and obese patients more often fulfilled the TTH criteria than normal-weight children [17,18,39].

The mechanism involved in migraine in obese children is not known and several hypotheses have been suggested. The pathways involved in feeding regulation and those implicated in migraine are consistent in many central and peripheral points, for example, in hypothalamic activation, in the release of adiponectin, serotonin, or other immune modulators and inflammatory neurotransmitters [38,44,45]. Other possible factors involved in the migraine–obesity relationship are lifestyle and habits including migraine medications that could modulate body weight as well as diet and physical exercise of patients [40,44,45,46,47].

### 3.2. Adiponectin

Adiponectin is a protein post-translationally modified into various multimers and secreted into the circulation from adipocytes [48]. It shows anti-inflammatory properties and also protects cells from apoptosis. Adiponectin receptors are expressed in the cortex, hypothalamus, brainstem, circumventricular organs, and on the endothelium of the cerebral microvasculature. Several signal transduction mechanisms were found to be activated by adiponectin, of which some are implicated in migraine including nuclear factor kappa beta (NFkβ), AMP-activated protein kinase (AMPK), mitogen-activated protein kinase (MAPK), or endothelial nitric oxide synthase (e-NOS) [49,50]. Previous research demonstrated that obese children had low adiponectin levels, whereas the markers of inflammation and proinflammatory cytokines were elevated [51]. The study by Asayama et al. showed that the levels of adiponectin in obese children were correlated inversely with the visceral adipose tissue area and their serum levels were decreased. Low levels of adiponectin are associated with platelet aggregation as well as proinflammatory cytokine release [52]. The above-mentioned mediators can trigger the following cascade of events: they affect frequency, severity, and duration of migraine attacks, which, especially when repeated, may cause central sensitization, and eventually permanent neuronal damage. Duarte et al. reported significantly higher levels of adiponectin in patients with migraine than in the controls [53]. However, no relation was observed when patients with episodic vs. chronic migraine as well as migraine patients with aura vs. without aura were compared [53]. On the other hand, a Brazilian study revealed a statistical difference in adiponectin levels between migraine patients and tension-type headaches [54]. The study by Dearborn et al. demonstrated that total adiponectin levels were significantly higher in women compared to men [55].

### 3.3. Dyslipoproteinemia

Dyslipoproteinemia, just like obesity, is a common, well-known risk factor for cerebrovascular and cardiovascular diseases [56,57]. Data from large groups of subjects revealed how common the problem of lipid disturbances in children is. In a study by Reuter et al., 42% (out of n = 1243) of healthy Brazilian children and adolescents had dyslipidemia, which was more prevalent in girls than in boys [15]. Almost 20% of Korean children and adolescents aged 10–18 years were reported to have at least one abnormal lipid profile [16].

Elevated lipid levels were observed in adults with idiopathic headaches, but there are only a few small clinical studies showing a relationship between disturbances in lipid levels and migraine in pediatric patients [14,22,58,59,60,61]. A significant positive correlation between the frequency and intensity of migraine and the levels of total cholesterol as well as low-density lipoprotein (LDL) cholesterol was observed in an Italian study. Moreover, in patients treated for migraine prophylaxis, a decreased number and intensity of episodes correlated significantly with a reduction in total cholesterol and LDL levels [14]. Glueck and Bates showed a similar relation between severe migraine headaches and LDL levels in boys with primary and familial dyslipoproteinemias [58]. In a large group of Brazilian adults, the relation between migraine and lipid sub-fraction was assessed [60]. The authors indicated a positive correlation between migraine without aura and the highest tertiles of very-low-density lipoprotein (VLDL) cholesterol in females. On the other hand, in men, a positive association was observed between migraine with aura and the highest tertile of VLDL_3_ cholesterol [60].

Nevertheless, some authors deny the correlation between lipid levels and the occurrence of primary headaches in children. A large and representative American sample showed no differences in the values of lipid levels by headache status. However, boys with headaches had lower high-density lipoprotein (HDL) levels than asymptomatic patients of the same sex [22].

In contrast, Winsvold et al. reported that total cholesterol levels may be even lower in adults suffering from migraine than in non-migraine controls. In this study, HDL cholesterol showed a similar tendency toward decreased values [61].

Moreover, according to Sacher et al., migraine correlates with an increased risk of ischemic stroke in adult patients in comparison to the healthy population [62,63]. Studies performed in children did not confirm this correlation, but some authors reported a higher prevalence of migraine among young individuals with medical history of stroke, silent infarct-like brain lesions, and cerebrovascular disorders during headache attacks [64,65,66]. The study by Sarecka-Hujar et al. demonstrated that about 40% of children with arterial ischemic stroke had hypertriglyceridemia [67]. Moreover, the data from the International Pediatric Stroke Study (IPSS) demonstrated that 41% of children with arterial ischemic stroke had elevated levels of TG, whereas 36% of children were dyslipidemic [68].

### 3.4. New Potential Markers

Recent data suggest new potential factors involved in primary headache pathophysiology. Some new potential risk factors for atherosclerosis and platelet aggregation (i.e., brain-derived neurotrophic factor (BDNF), sCD40L (soluble CD40 ligand), serpin E1/PAI I (endothelial plasminogen activator inhibitor) and vascular endothelial growth factor (VEGF)) were earlier correlated with headaches [21,69,70,71,72,73,74,75,76,77]. The role of these factors seems controversial since the results of clinical studies are often inconsistent. However, exploring new ideas about headache pathophysiology gives additional treatment opportunities, especially in medical prognosis and prevention of vascular risk factors. In this section, we have discussed the results concerning the above-mentioned markers, which we believe are the most interesting and the most relevant to the topic.

#### 3.4.1. Brain-Derived Neurotrophic Factor

BDNF is also one of the neurotrophins taking part in pain generation and modulation in patients with migraine [23,24]. BDNF plays an important role in the nociceptive pathway and contributes to central sensitization. Its action is dose-dependent: low doses lead to hyperalgesia and high ones result in analgesia. It co-expresses together with CGRP (calcitonin gene-related peptide), another important biomolecule of migraine, in trigeminal ganglion neurons [78,79,80,81].

Its level is significantly elevated during the ictal period of migraine [24,25]. According to the results of thee meta-analyses of over 1000 patients suffering from migraine, BDNF gene polymorphisms, rs6265 and rs2049046, are associated with common migraine in the Caucasian population; both data came from adult migraineurs [82,83]. Our preliminary data demonstrated higher levels of BDNF in children with headaches than in the controls [59]. Presumably, the elevated BDNF level may result from hypoxia, which induces its production. Previously, it was observed that BDNF plays a beneficial role in the brain as it may promote neurite outgrowth as well as protect neurons from apoptosis induced by serum deprivation [84].

Chronic tension-type headaches (CTTH), concerning nearly 3% of the general population, are the most common headaches all over the world, both in children and adults. The data from adults suffering from CTTH suggest that BDNF may be a serum marker of neuroplasticity induced by electroacupuncture [85]. Chassot et al. studied the relationship between the electroacupuncture effect and serum BDNF levels in adults with CTTH. The authors noted that the elevation of the marker levels after electroacupuncture was inversely proportional to the perceived pain. Interestingly, the electroacupuncture effect on both pain and BDNF persisted beyond the active intervention period [85]. The mechanisms underlying this phenomenon are not fully understood, but they may result from the modulatory multidirectional role of BDNF in neurogenesis, pain pathophysiology, and synapsis functioning that was previously reported in brain ischemia [86,87]. BDNF is involved in processes maintaining the homeostasis of intracellular calcium and regulating the expression of the cell membrane calcium channel proteins [88]. It may reduce the negative influence of free radicals on cells by downregulating the expression of superoxide dismutase [89]. Glutamate excitotoxicity may also be limited by the downregulation of BDNF on NMDA receptors [90]. Moreover, its impact on sensitization processes may also take place at the level of neuronal genes. The agents involved in the gene expression are upstream element-like transcriptional factors c-jun and c-fos, which are regulated by BDNF [91]. The authors did not find data specifically related to episodic TTH.

#### 3.4.2. Soluble CD40 Ligand

sCD40L is a ligand of the glycoprotein IIb-IIIa receptor; its role is to stabilize the thrombus and platelet activation. It is also the predictor of cardiovascular diseases and early atherosclerosis [92,93,94]. sCD40L belongs to tumor necrosis factors and is expressed on the surface of hematopoietic, endothelial, and smooth muscle cells [26]. Activated platelets produce sCD40L, which contributes to plaque instability [95,96,97].

According to the original data by Guldiken et al., elevated levels of sCD40L are found in adult migraine patients in the interictal period, which shows not only the connection between migraine and vascular disorders, but is also perceived as the risk factor for cardiovascular diseases (myocardial infarction, stroke) [26]. The high concentration of sCD40L in migraine patients was the result of platelet activation such as aggregation and secretion of thromboxane A2, thrombomodulin, and platelet factor 4 [26,95,96,97]. sCD40L stimulates the production of proinflammatory cytokines (IL-1, IL-6, IL-8, IL-10, TNF), which are pain mediators and are suggested to be implicated in neurovascular inflammation resulting in migraine pain [27,98]. The study of Domingues et al. showed increased levels of IL-8 as well as monocyte chemoattractant protein-1 in patients with tension-type headaches [54]. A study based on migraine patients from China demonstrated that they had higher levels of IL-6, IL-1β, and TNF compared to healthy controls [27]. The authors found the level of calcitonin gene-related peptide to be significantly correlated with the level of IL-1β and IL-6.

#### 3.4.3. Inhibitor of Plasminogen Activator-1

PAI-1 is the main inhibitor of fibrinolysis and is considered to be a marker of thromboembolic disorders. PAI-1 is a 48 kDa glycoprotein belonging to the serine protease inhibitor family (SERPIN) consisting of 379 amino acids [98]. PAI-1 is secreted, among other, by endothelial cells, hepatocytes, platelets, and smooth muscle cells. It is a major factor modulating the activity of plasminogen activators of both tissue (t-PA) and urokinase (u-PA) type. It forms a stable complex with plasminogen activators, which is then removed from the circulation by hepatocytes. PAI-1 is present in both active and inactive form. The active form of PAI-1 is unstable, with the half-life of 30 min [99]. PAI-1 secretion by different cell types is induced and regulated by various factors including IL-1 and IL-6, TNF-α, TGFβ (transforming growth factor β), EGF, fibrin esters, insulin, and lipoprotein of VLDL fraction [100,101,102,103]. Some of these factors exert a direct influence on the transcription of the PAI-1 gene. It appears that the physiological properties of PAI-1 are, at least in part, the result of changes in the transcriptional rate of the gene encoding the protein, and certain elements of response (e.g., to glucocorticoids), which are present within the gene promoter, control these properties.

Elevated plasma levels of PAI-1 are associated with a variety of thromboembolic disorders [104,105]. It was also considered as an independent risk factor for recurrent myocardial infarction in patients who had the first event before the age of 45 [106]. The results of some studies indicate that elevated levels of PAI-1 are associated with ischemic heart disease progression in young men with a history of myocardial infarction as well as the first episode of myocardial ischemia [106,107,108].

There are few data concerning the role of plasma PAI-1 levels in the course of headaches. Previously, reduced levels of PAI-1 as well as t-PA were observed in migraineurs. However, the data should be analyzed with caution due to the low number of patients [28]. In the study by Kes et al., patients with migraine had a high frequency of homozygotes and heterozygotes for PAI-1 polymorphism compared to the controls [109].

Hypoxia is one of the mechanisms related to migraine. Enhanced expression of PAI-1 may be caused by arterial oxygen desaturation, followed by decreased expression of plasminogen activators, which cause blood hypercoagulability and paradoxical microembolism [29,110,111].

#### 3.4.4. Vascular Endothelial Growth Factor

VEGF is viewed as an important angiogenesis, vasculogenesis, and tissue remodeling factor [112]. VEGF affects angiogenesis, observed in some pathological conditions such as diabetic retinopathy, cancer, rheumatoid arthritis, or psoriasis [113,114]. VEGF is a glycosylated homodimer with a molecular weight of 46–48 kDa, dissociating into two monomers connected by disulfide bonds [101]. The VEGFR2 receptor is the main receptor for VEGF, by which it induces a number of biological responses.

Hypoxia is one of the main factors inducing mRNA expression for VEGF. In response to low oxygen partial pressure, the accumulation of adenosine may occur, causing an increase in cAMP concentration, which leads to an increase in the level of VEGF mRNA [114]. In addition, growth factors, various types of inflammatory cytokines, or mutations of some oncogenes leading to neoplastic transformation may also stimulate VEGF expression.

Nevertheless, some recent studies suggest its role in energy hemostasis as well as its influence on adipose tissue function. The increased expression of VEGF is believed to protect against insulin resistance and diet-induced obesity [112].

The loss of a single VEGF allele may even be lethal as it leads to serious disturbances in vascular development and cardiovascular defects [115]. The association between VEGF gene polymorphism and the occurrence of diseases is also an area of interest in headache studies. Gonçalves et al. examined the association between three functional polymorphisms in the promoter region (-2578C>A, -1154G>A, -634G>C) and VEGF haplotypes in migraineur women compared with asymptomatic controls. The authors revealed no differences in the distribution of VEGF genotypes and alleles between the groups. However, the more frequent occurrence of the AGC haplotype in subjects with migraine with aura compared to healthy subjects may indicate that VEGF haplotypes play a role in idiopathic headache susceptibility [116].

Michalak et al. [117] found that VEGF levels in serum were decreased during the interictal period in young Polish adults with migraine compared to the controls. Simultaneously, the authors also demonstrated higher levels of total cholesterol, LDL-cholesterol, and triglycerides in patients with migraine compared to healthy subjects [117]. Previously, Blann et al. reported that VEGF levels were significantly reduced in patients with hyperlipidemia and without atherosclerosis after lipid-lowering therapy [118]. On the other hand, VEGF levels were found to be significantly increased in Spanish patients with migraine compared to the control subjects [30]. A study based on a group of Polish children with headache indicated no difference in VEGF levels between children with headache and controls. However, boys with headache had higher VEGF levels than girls [59].

## 4. Conclusions

Headache is one of the most common neurologic conditions in pediatric clinical practices. Current headache classification is based on the International Classification of Headache Disorders, 3^rd^ edition, which defines four main categories of primary headaches: migraine, tension-type headaches, trigeminal autonomic cephalalgias, and other primary headache disorders. The medical history, physical, and neurological examination play a significant role in headache diagnosis. Treatment of headaches in children requires a balanced approach to personalize therapy depending on the kind, frequency, and severity of symptoms, and includes three general stages: lifestyle modification, psychological therapy, and pharmacotherapy. There are new potential risk factors involved in headache pathophysiology. Pediatric studies suggest that obesity is associated with headache disorders. Many authors report that the risk of migraine increases with increasing obesity status. Disturbances in lipid blood levels were observed in adults and pediatric migraineurs. Headaches were observed when total cholesterol levels went up and down beyond normal ranges. Studies performed in children did not confirm the hypothesis that headaches are associated with a higher risk of ischemic stroke. However, migraine may be one of the manifestations of vascular disturbances including atherosclerosis in pediatric patients. This is why migraine onset in pediatric age should be an indication for blood lipid evaluation. Recent data suggest new potential risk factors for atherosclerosis and platelet aggregation such as adiponectin, BDNF, sCD40L, TIMP1, serpin E1/PAI I, and VEGF. The differences in blood levels of these biomarkers in young individuals with headaches compared to healthy controls have been recently reported by many authors. Nevertheless, there are methodological limitations and contradictory results of the studies in this area. Further research on the association between headache disorders and vascular changes in children is needed to clarify this issue.

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
