# Peer review of "Headache in Children: Selected Factors of Vascular Changes Involved in Underlying Processes of Idiopathic Headaches"

_children, 2020, doi:10.3390/children7100167_

Round 1

Reviewer 1 Report

In my opinion, the present review of the literature is an interesting topic research that has been subject of interest in recent years, although I personally think that way in which the potential markers of atherosclerosis and platelet aggregation have been managed in the present review is not pertinent and not add relevant information for the main issue, headache in children. Overall, I think it need to be improved. Here are my main concerns and comments:

Firstly, in my opinion the tittle of manuscript could confuse readers. The tittle of the present study is “Headache in children: a great challenge in everyday practice: classification, diagnosis, treatment and new potential factors involved in pathophysiology”. As authors specify between lines 53 and 54 the main study aim is focused in classification, diagnosis and treatment of headaches, although in tittle author reflect that new potential factors involved in pathophysiology will be addressed in the manuscript, but in the text authors do not address factor involve in headache pathophysiology, authors only have exposed the of the potential markers of atherosclerosis and platelet aggregation and the role they play in certain processes, but not the role they play in headache

Is this review registered in PROSPERO? If it has been registered authors must provide the register number of the review in PROSPERO.

Authors have expressed in lines 56-57 that the evidence available in this field is scarce. This kind of reviews require of an important research effort. In my opinion, author should improve their research strategy, it seem to be insufficient for this kind of study. Why have authors not searched in others electronic data bases such as, Scopus, CINAHL, SciELO or Web of Science? Why authors not employ standardized terms, for example MeSH term in PubMed such as, “pediatrics”, “child”, “migraine disorders”, tension-type headache? These advices could improve the research strategy, being more accessible more recent studies in this research field.

Bibliography is old-fashioned. I have observed that, surprisingly, less of the 14% of the 104 references are more recent than 7 years. Why authors have not implemented the last classification of International classification of headache disorders III edition, developed in 2018 (Headache Classification Committee of the International Headache Society (IHS). The International Classification of Headache Disorders, 3rd edition. 2018;38:1-211. DOI: 10.1177/0333102417738202.)?? I recommended to implement it in the present manuscript. In addition I recommend authors to take into account the following references:

  • GBD 2016 Headache Collaborators. Global, regional, and national burden of migraine and tension-type headache, 1990–2016: a systematic analysis for the Global Burden of Disease Study 2016. Lancet Neurol. 2018;17:954-976. DOI: 10.1016/S1474-4422(18)30322-3.
  • Practice guideline update summary: Acute treatment of migraine in children and adolescents: Report of the Guideline Development, Dissemination, and Implementation Subcommittee of the American Academy of Neurology and the American Headache Society. 2020;94(1):50.doi: 10.1212/WNL.0000000000008728.
  • Rizzoli P, Mullally WJ. Headache. Am J Med. 2018;131:17-24. DOI: 10.1016/j.amjmed.2017.09.005.
  • Ciçek Wöber-Bingöl. Epidemiology of migraine and headache in children and adolescents. Curr Pain Headache Rep. 2013 Jun;17(6):341. doi: 10.1007/s11916-013-0341-z.
  • William P Whitehouse1 2, Shakti Agrawal. Management of children and young people with headache. Arch Dis Child Educ Pract Ed. 2017 Apr;102(2):58-65. doi: 10.1136/archdischild-2016-311803.
  • Jasmin M Dao, William Qubty. Headache Diagnosis in Children and Adolescents. Curr Pain Headache Rep. 2018 Feb 23;22(3):17. doi: 10.1007/s11916-018-0675-7.

Author Response

Dear Reviewer,

we would like to thank you for Your review of our manuscript. We appreciate Your corrections and suggestions that ensued. A major revision of the paper has been carried out to take all of them into account. The linguistic correction of the manuscript was also performed.

Below Authors present the responses and comments to the reviews.
We hope our article has been significantly improved and meets Your expectations.

Yours sincerely,

Authors

The responses and comments to the reviews

Firstly, in my opinion the tittle of manuscript could confuse readers. The tittle of the present study is “Headache in children: a great challenge in everyday practice: classification, diagnosis, treatment and new potential factors involved in pathophysiology”. As authors specify between lines 53 and 54 the main study aim is focused in classification, diagnosis and treatment of headaches, although in tittle author reflect that new potential factors involved in pathophysiology will be addressed in the manuscript, but in the text authors do not address factor involve in headache pathophysiology, authors only have exposed the potential markers of atherosclerosis and platelet aggregation and the role they play in certain processes, but not the role they play in headache
The manuscript has been reorganized in accordance with the reviewers' suggestions to focus on the most important issues related with atherosclerosis and platelet aggregation in children with primary headaches.

The tittle has also been changed to better reflect the content of the work. In the title we have included only the potential factors underlying the processes related to atherosclerosis and platelet aggregation, not headache pathophysiology. We hope that the title will now reflect the content of the work and be more useful for the readers.

Is this review registered in PROSPERO?
No. This review is not registered.

Authors have expressed in lines 56-57 that the evidence available in this field is scarce. This kind of reviews require of an important research effort. In my opinion, author should improve their research strategy, it seems to be insufficient for this kind of study. Why have authors not searched in others electronic data bases such as, Scopus, CINAHL, SciELO or Web of Science? Why authors not employ standardized terms, for example MeSH term in PubMed such as, “pediatrics”, “child”, “migraine disorders”, tension-type headache? These advices could improve the research strategy, being more accessible more recent studies in this research field
Bibliography is old-fashioned. I have observed that, surprisingly, less of the 14% of the 104 references are more recent than 7 years. Why authors have not implemented the last classification of International classification of headache disorders III edition, developed in 2018 (Headache Classification Committee of the International Headache Society (IHS). The International Classification of Headache Disorders, 3rd edition. 2018;38:1-211. DOI: 10.1177/0333102417738202.)?? I recommended to implement it in the present manuscript.

Authors decided to use also old-fashioned publications in some paragraphs because of their substantive value. There are often no good recent publications on the issues raised in children – especially epidemiology. However, Authors are aware of the importance of the current bibliography and we have made every effort to enrich the bibliography with recent publications. Additional databases and standardized terms were used to obtain current data. The last classification of International Classification of Headache Disorders III edition, developed in 2018 has been implemented.
We would like to thank You for this suggestion. We believe that completing the bibliography has made our manuscript more up to date and valuable.

The Reviewer recommended to take into account some references.
Thank you for suggesting some new publications. Authors considered these papers valuable and included them in the manuscript.

Reviewer 2 Report

Overall Comments

  1. The sections go from Introduction (1) to Methodology (2) and then to different results which should all be labeled under a separate section (3) before Conclusions (currently 6 but should be 4). This would be very helpful in the organization of the paper
  2. I am a bit confused as to the overall intent of the paper. I don’t think there is a large need in the literature to go over the diagnosis and treatment of headaches in children currently, and I personally would still find your paper to be useful focusing on the updates on atherosclerosis and lipids with migraine, as well as the other biomarkers
  3. Throughout most of the paper the authors should carefully go through and insert references where appropriate. I have pointed out a few but this is a theme that is present throughout this paper.

Abstract and Title

  1. The abstract jumps around a bit starting with line 24 (Migraine was found to be…)
  2. It seems from the abstract that the paper mainly focuses on the link between migraine and lipids and atherosclerosis. The abstract should reflect this a bit more as it currently suggests a broader review of pathophysiology in migraine

Introduction

  1. At the beginning there are statistics quoted with overall prevalence of headache, however the discussion regarding odds ratios only applies to kids with migraines. The discussion of prevalence in kids should probably focus a bit more on migraine (possibly after giving some of those over-arching statistics)
  2. It may be helpful to give some background information on prevalence of lipid metabolism disorders and/or atherosclerosis in kids
  3. I would consider deleting sections 1.1-1.3
  4. Section 1.4 needs references for each statement
    1. Shares a reference with page 2 line 49 regarding migraine being suggested as the first sign of disorders of lipid metabolism

Results (sections 3-5)

  1. BMI
    1. Page 8 in the first paragraph it would be helpful to discuss what kind of reductions in weight were associated with improvements in headache (did patients see a benefit after losing a certain amount of weight or BMI change?)
    2. There is a good amount of information regarding adiponectin and migraines that would be prudent to expand on
    3. The last 2 sentences are unclear as to their meanings
  2. Dyslipoproteinemia
    1. Line 247- it may be helpful to discuss if the association is with high levels of LDL or low, and same with HDL
    2. This section jumps around with a lot of conflicting studies, so reorganizing this may be helpful for the reader to make some sense of this.
    3. Again, there are many statements used that are not supported by references, and when these statements are made to contrast against other statements that do have references this is confusing.
  3. New Potential Markers
    1. 1
      1. Is there any data that supports that BDNF contributes to the pathophysiology of migraine, otherwise we aren’t sure if the BDNF has any role or is it just the CGRP (both are elevated, but we know when CGRP is blocked this alleviates migraine.
      2. The last paragraph should be expanded to give a better understanding of the potential mechanisms in CTTH, as well as any data from episodic TTH
    2. 2
      1. This section also jumps around between migraine and TTH, and could benefit from reorganization
    3. 3
      1. Page 9 line 308- consider removing the sentence regarding vitronectin as it doesn’t really add much to the role of PAI-1 in headaches and is a bit of a tangent in the discussion
    4. 4
      1. In your discussion of reference 103 it is unclear what groups were being compared

Conclusion

  1. You seem to look at atherosclerosis as a cause of secondary headache, rather than association with migraine. This is not clearly discussed at all in the rest of the paper.  I think it is an important point, but should be clearly supported by the evidence.

Author Response

Dear Reviewer,

we would like to thank you for Your review of our manuscript. We appreciate Your corrections and suggestions that ensued. A major revision of the paper has been carried out to take all of them into account. The linguistic correction of the manuscript was also performed.

Below Authors present the responses and comments to the reviews.
We hope our article has been significantly improved and meets Your expectations.

 Yours sincerely,

Authors

 The responses and comments to the reviews

  1. The sections go from Introduction (1) to Methodology (2) and then to different results which should all be labeled under a separate section (3) before Conclusions (currently 6 but should be 4). This would be very helpful in the organization of the paper

Authors admit that the organization of the manuscript was unclear and could confuse readers. The organization of the manuscript has been changed to make it clearer. We have divided our text into three parts: Introduction, Methodology, Result and Conclusion. We believe that this change will make the text more useful.

  1. The paper would be more useful if it was focused on the updates on atherosclerosis and lipids with migraine, as well as the other biomarkers.

I would consider deleting sections 1.1-1.3

Initially Authors wanted to go over the general issues such as classification, diagnosis and treatment to make the text more friendly and practical for the doctors of various specialties – not only the neurologist.

However, we decided to delete this part of the Introduction as suggested to focus and analyze in depth the main issue of the manuscript – the relation between idiopathic headaches and the potential markers of atherosclerosis and platelet aggregation. We hope that our text is now more professional and up-to-date.

  1. Throughout most of the paper the authors should carefully go through and insert references where appropriate.
    Authors regret the deficiencies in the references. These mistakes probably arose while editing the text. The references have been checked and inserted in the appropriate places.

  1. The abstract jumps around a bit starting with line 24 (Migraine was found to be…)
    It seems from the abstract that the paper mainly focuses on the link between migraine and lipids and atherosclerosis. The abstract should reflect this a bit more as it currently suggests a broader review of pathophysiology in migraine.
    The abstract has been revised to more accurately describe the actual content of the article and to reflect the more broad link between different risk factors of headache and it's pathophysiology.

  2. At the beginning there are statistics quoted with overall prevalence of headache, however the discussion regarding odds ratios only applies to kids with migraines. The discussion of prevalence in kids should probably focus a bit more on migraine (possibly after giving some of those over-arching statistics)

As suggested, authors have made efforts to extended and supplemented the data both in the Introduction and in the Discussion. After correction made on the Reviewer's suggestions the presented data reflect the statistics both in headache in general, and specifically- in migraine. We hope that the text now provides a wide range of information on this subject.

  1. It may be helpful to give some background information on prevalence of lipid metabolism disorders and/or atherosclerosis in kids.

Thank You very much for the suggestion. Indeed, the problem of dyslipidemia in pediatric population is wide spread and is considered to be a risk factor of many health problems especially these of vascular origin. In primary headache in children lipid disorders are mostly considered together with other factors like obesity and environmental factors influence.  New up-to-date data on this issue have been presented in the manuscript. Authors have added information obtained from studies performed on large populations to give a broader view of the issue.

  1. Section 1.4 needs references for each statement
    1. Shares a reference with page 2 line 49 regarding migraine being suggested as the first sign of disorders of lipid metabolism

As we wrote earlier, the references have been checked and inserted in the appropriate places.

  1. Page 8 in the first paragraph it would be helpful to discuss what kind of reductions in weight were associated with improvements in headache (did patients see a benefit after losing a certain amount of weight or BMI change?)

Thank You for this comment. This issue does seem relevant and interesting. This part of the manuscript has been expanded to include missing information based on the recent data from the weight loss program in obese adolescents with migraine. We would like to underline that the data on this matter are lacking in up-to-date literature so there is no unequivocal answer for the question stated above. 

  1. There is a good amount of information regarding adiponectin and migraines that would be prudent to expand on.

We agree that including information about the role of adiponectin in the headaches makes our manuscript more valuable. Authors have prepared a part describing the potential role of the adiponectin in headaches used current scientific reports. We are convinced that this information significantly increased the value of our manuscript. However, we must note that data on such relations in children are very limited.

  1. Dyslipoproteinemia
    1. Line 247- it may be helpful to discuss if the association is with high levels of LDL or low, and same with HDL
    2. This section jumps around with a lot of conflicting studies, so reorganizing this may be helpful for the reader to make some sense of this.
    3. Again, there are many statements used that are not supported by references, and when these statements are made to contrast against other statements that do have references this is confusing.

Thank You very much for the remark. We followed Your suggestion carefully;  this paragraph has been reorganized and enriched with additional information about lipid levels in primary headaches. We have made every effort to get as many up- to- date publications as possible to present a thorough knowledge of this topic.
The references have been checked and inserted in the appropriate places

  1. New Potential Markers
  1. Is there any data that supports that BDNF contributes to the pathophysiology of migraine, otherwise we aren’t sure if the BDNF has any role or is it just the CGRP (both are elevated, but we know when CGRP is blocked this alleviates migraine)

Thank you for pointing out this inaccuracy. Authors supplemented the information on associations of BDNF with migraine and potential mechanisms underlying this phenomenon. Once again- the BDNF is being considered to be a risk factor for endothelial dysfunction and the diseases coming from the problem, still- it's role in pathogenesis of pediatric cerebrovascular disorders is the matter of research and needs more thorough evaluation.

    1. The last paragraph should be expanded to give a better understanding of the potential mechanisms in CTTH, as well as any data from episodic TTH

Thank You very much for the suggestion. Indeed, the problem of TTH etiology and risk factors especially in the aspect of its episodic appearance is very interesting and worth to be thoroughly checked. We have expanded this paragraph to include missing information. We hope this point will be clear to the readers now. However, Authors did not find data specifically related to episodic TTH – what was marked in the text.

    1. This section also jumps around between migraine and TTH, and could benefit from reorganization

Thank You once again for the remark making our paper more clear and comprehensive. As suggested, the paragraph concerning new potential markers has been reorganized and expanded. We tried to deepen the information about the role of potential factors of atherosclerosis and platelet aggregation in primary headaches. Although data on children are sparse and often contradictory, we hope that this paragraph is now clear and valuable.

    1. Page 9 line 308- consider removing the sentence regarding vitronectin as it doesn’t really add much to the role of PAI-1 in headaches and is a bit of a tangent in the discussion.

Authors agree that this sentence adds nothing new to the manuscript. This sentence has been deleted.

    1. In your discussion of reference 103 it is unclear what groups were being compared

Thank you for pointing out this ambiguity. Authors have added a detailed description of the study groups and the purpose of the study. In our opinion this work raises important and interesting issues related to VEGF gene polymorphism that may be useful for the readers.

  1. Conclusion

You seem to look at atherosclerosis as a cause of secondary headache, rather than association with migraine. This is not clearly discussed at all in the rest of the paper.  I think it is an important point, but should be clearly supported by the evidence.

Thank You for this comment. The aim of our review was to present current knowledge on the role of potential markers of atherosclerosis and platelet aggregation which may be associated with primary headaches. This paragraph has been reorganized to avoid confusing the reader with additional thread on secondary headaches that is not clearly supported by the literature in the manuscript.

Round 2

Reviewer 1 Report

It is possible to appreciate a remarkable effort to improve the manuscript. However, in my opinion, methods section should be improved in order to facilitate reader understanding:

  • In lines 228-229, authors claim that their research strategy was “the typical methodology used in systematic reviews”. In my opinion this is not the best way to describe their research strategy. This description is ambiguous, not very detailed and inappropriate for a document of scientific relevance, which should ensure the reproducibility of the results obtained. I suggest author to include in this section the research strategy used for at least one of the electronic data bases employed.
  • The eligibility criteria are poorly described. Exclusion criteria do not appear in methods section and inclusion criteria must be described deeply.

Reviewer 2 Report

I think that the changes you have made to the paper have made it much more cohesive and definitely interesting.  I still think that this paper would benefit from significant English-language proofreading prior to publication, however.
